# Tasks and Experiences of the Prospective, Longitudinal, Multicenter MoMar (Molecular Markers) Study for the Early Detection of Mesothelioma in Individuals Formerly Exposed to Asbestos Using Liquid Biopsies

**DOI:** 10.3390/cancers15245896

**Published:** 2023-12-18

**Authors:** Daniel Gilbert Weber, Swaantje Casjens, Katharina Wichert, Martin Lehnert, Dirk Taeger, Hans-Peter Rihs, Thomas Brüning, Georg Johnen

**Affiliations:** Institute for Prevention and Occupational Medicine of the German Social Accident Insurance (IPA), Ruhr University Bochum, 44801 Bochum, Germany

**Keywords:** asbestos, biomarker, blood-based, calretinin, cancer, cohort, liquid biopsies, lung cancer, mesothelin, mesothelioma

## Abstract

**Simple Summary:**

Mesothelioma is an aggressive cancer, commonly detected at late stages. Proper blood-based biomarkers for the detection of the disease at early stages need to be assessed in prospective, longitudinal studies. This article comprehensively elucidates underlying methodologies to support researchers conducting these required studies.

**Abstract:**

Mesothelioma is an aggressive cancer, strongly associated with prior exposure to asbestos. Commonly, tumors are detected at late stages of the disease. Detection at early stages might be meaningful, because therapies might be more effective when the tumor burden is relatively low and the tumor has not spread to distant sites. Circulating biomarkers in blood might be a promising tool to improve the early detection of mesothelioma, but for screening in asymptomatic subjects, candidate biomarkers need to be validated in appropriate studies. This study was conducted to assess the performance of biomarkers in liquid biopsies to detect mesothelioma at early stages. Over a period of 10 years, 2769 volunteers formerly exposed to asbestos were annually examined and liquid biopsies were collected. A follow-up was completed 17 months after the last blood collection. The article provides a detailed overview of our lessons learned and experiences of conducting a prospective, longitudinal, multicenter study. The existing cohort of individuals at risk is highly suitable for the validation of blood-based biomarkers for the early detection of mesothelioma as well as lung cancer.

## 1. Introduction

Mesothelioma is an aggressive cancer of the serous membranes commonly associated with prior exposure to asbestos. In Germany, on average, 1279 men and 337 women were diagnosed with mesothelioma annually between 2010 and 2019 [1]. The majority of the cases were considered to be caused by occupational hazards. Worldwide, 30,870 new cases and 26,278 deaths were related to mesothelioma in 2020, representing 0.2% and 0.3% of all cancer cases and deaths, respectively [2].

Mesothelioma is characterized by a long latency period of up to 50 years and epithelioid mesothelioma is the most common subtype (69%), whereas biphasic and sarcomatoid mesotheliomas account for 12% and 19% of all cases, respectively [3]. Usually, mesothelioma is diagnosed at late stages of the disease and the survival time after diagnosis is between six and nine months [4]. However, the epithelioid subtype shows a slightly better prognosis in comparison to that of the sarcomatoid and biphasic subtypes. Therapy with pemetrexed and cisplatin showed an increase in survival to 12.1 months (with a 95% confidence interval (CI) of 10.0–14.4 months) [5]. This combination of chemotherapeutics has been the only approved first-line treatment in the past two decades. Recently, the use of nivolumab (a human anti-programmed cell death protein 1 (PD-1) antibody) and ipilimumab (a human anti-cytotoxic T-lymphocytes-associated protein 4 (CTLA-4) antibody) showed clinical benefits for mesothelioma patients with an extended survival of 18.1 months (with a 95% CI of 16.8–21.4 months) [6]. However, the benefit was larger for patients with non-epithelioid mesothelioma in comparison to that of those with epithelioid mesothelioma. In 2020, the combination of nivolumab and ipilimumab was approved as a first-line treatment for unresectable pleura mesothelioma.

A central goal of oncology research is the use of minimally invasive methods to assess and detect cancer early [7]. Liquid biopsies might be a promising tool for the early detection of tumors, because it is not possible to collect samples from asymptomatic subjects by an invasive procedure on the target organ [8]. In contrast to invasive methods, liquid biopsies comprise the detection of molecular biomarkers in easily accessible body fluids such as blood, the most common matrix for biomarker evaluation [9]. Circulating proteins, DNA, e.g., DNA methylation status, and RNA, e.g., microRNAs (miRNAs), in the blood represent the most promising sources of candidate biomarkers. Appropriate biomarkers should be easy to detect, robust against influencing factors, and marked by sufficiently high levels of sensitivity and specificity [8]. Thus, biomarkers might be a promising tool for screening in asymptomatic subjects in order to detect cancer at early stages, because therapies might be more effective when the tumor burden is relatively low and the tumor has not spread to distant sites [7].

A common strategy to establish novel diagnostic biomarkers consists of two stages: discovery and verification. The discovery stage includes the identification of promising candidate biomarkers and the subsequent verification stage includes the assessment of the biomarkers’ performance. Both parts are usually performed in cross-sectional comparisons of symptomatic cases and controls without the target disease. However, for the early detection of cancer, the candidate biomarkers need to be additionally validated in a third stage, utilizing prospective, longitudinal studies with serial sampling in order to assess the potential of the candidate biomarkers to detect the aimed disease prior to the clinical diagnosis when symptoms have not occurred yet. In general, cross-sectional studies are not suitable for this purpose, because cases with already diagnosed diseases are included. In addition, these cases are usually at later stages of the disease, whereas appropriate biomarkers for early detection should indicate a disease before the occurrence of clinical symptoms [8]. Thus, biomarkers for early detection must be validated in prediagnostic samples. For the assessment of the candidate biomarkers to detect a disease at early stages, a prospective cohort of individuals at risk is essential.

This article provides an overview of our lessons learned and experiences of conducting the MoMar (Molecular Markers) study on incident mesothelioma to support researchers performing prospective, longitudinal, multicenter studies in order to validate biomarkers for the early detection of cancer.

## 2. Tasks for Conducting the Study

### 2.1. Definition of the Prospective Cohort

A valuable screening group consists of individuals at risk who are more likely to benefit from screening [10]. Thus, all participants to be recruited for the MoMar study were formerly exposed to asbestos. Besides mesothelioma, exposure to asbestos is associated with lung cancer as well as benign diseases, which might also be fatal [11].

In Germany, workers formerly exposed to asbestos are offered a surveillance program by the German Social Accident Insurance (Deutsche Gesetzliche Unfallversicherung (DGUV)), including regular medical examinations and X-ray or computer tomography (CT) scans of the thorax at various intervals between one and three years. A similar program is offered by individual statutory accident insurances (which are part of the DGUV) to asbestos workers with recognized benign occupational asbestos-associated diseases like asbestosis, pleural effusions, and pleural plaques, i.e., without mesothelioma or lung cancer. Taking advantage of the existing infrastructure, the participants of the MoMar study were recruited from the pool of participants of this surveillance program.

In a previous study conducted with a surveillance cohort of male German workers formerly exposed to asbestos, subjects with asbestos-related occupational diseases were associated with an increased relative risk (rate ratio = 6.0; 95% CI: 2.4–14.7) of developing mesothelioma [12]. Based on the increased risk and an initial assumption of 10,000 person-years, 38 mesothelioma cases (95% CI: 21–63 mesothelioma cases) were estimated to occur within the MoMar cohort, representing sufficient power to assess the performance of candidate biomarkers for the early detection of mesothelioma.

To reach 10,000 person-years, we aimed to recruit 2000 individuals with annual blood sampling over a period of five years. Besides recognized asbestos-associated diseases, no further inclusion or exclusion criteria were defined in order to establish a representative at-risk population. The advantage of such a prospective cohort is the clear definition of the target study population [13].

### 2.2. Selection of Participating Study Centers

Based on the targeted number of 2000 participants, the MoMar study had to be multicentered. Thus, it was necessary to set up a sufficient number of medical practices as collaborating study centers for the prospective, longitudinal study. As the surveillance program of the statutory accident insurances is offered via physicians to the individuals concerned, a high number of potential study centers exist in Germany.

Finally, a total of 26 study centers, which agreed to cooperate over the complete study period, were included. Regions meeting the following criteria were selected: those with increased incidence rates of mesothelioma, a number of patients with mesothelioma being routinely examined by the physicians, spatial proximity to the central study center in order to ensure the secure transport of the samples, and the support of the staff. Thus, the majority of study centers were located in Northern Germany due to its former shipbuilding industry (Figure 1).

All participating study centers obtained training by a field team and were supplied with all necessary study materials, i.e., ready-made bags with all required materials for blood collection and data recording, as well as centrifuges for immediate processing, and freezers for short-term storage of the samples at −20 °C in order to reduce the effort required by the staff of the participating study centers. The samples and corresponding documents were regularly collected by the field team and transported to the central study center.

The materials for blood collection and data recording were barcoded in advance for pseudonymization to ensure data protection and patient confidentiality as well as to avoid transfer errors. For pseudonymization, trustees were installed, enabling a secure follow-up of the participants during the prospective study. Thereby, samples were analyzed with the researchers blinded to the disease status according to good laboratory practice in biomarker research.

### 2.3. Selection of Appropriate Samples

The liquid biopsy concept supports the early detection of cancer [14]. Blood is a valuable matrix based on the general detectability of biomarkers in this medium and the well-established collection procedures in clinics and medical practices without a particular risk to the participants. However, a principal limitation of blood samples might be the risk of altered analytes in vitro [15]. The establishment of an appropriate sample collection procedure aiming at the best possible preservation of biomarkers via controlled preanalytical supervision should therefore be taken into account [9]. Accordingly, the sample collection procedure, the sample processing after collection, and the storage procedure need to be standardized in all participating study centers using predefined standard operating procedures (SOPs). The blood collection procedure is commonly assumed to be an easy part of clinical studies, but the collection and processing of the blood samples might have a meaningful impact on the general quality of the samples as well as on the quantity of the selected biomarkers, influencing the results of the study [15]. In multicenter studies, feasible SOPs have to be developed which should be easily implemented in the daily routine of the participating study centers. However, in multicenter studies, it is a challenge to ensure that all participating centers exactly follow the procedures as defined in the SOPs. As a consequence, compromises need to be made between perfect preanalytical processing and the reality of daily routine work [15].

Several preanalytical factors with impact on the samples prior to the analyses exist, e.g., storage temperature and freeze/thaw cycles. Additionally, common preanalytical problems include the inadequate quality of plasma samples, such as those with hemolysis and clotting [16]. Hemolysis in particular is the leading cause of unsuitable samples in laboratory practice, accounting for 40–70% of dropouts [16]. Regarding the analysis of biomarkers, it was shown, for example, that the grade of hemolysis influences the levels of miRNA in plasma [17]. Thus, appropriate quality controls have to be included in reliable studies in order to minimize the effect of preanalytical factors.

## 3. Experiences of Conducting the Study

### 3.1. Characteristics of the Study Cohort

Enrollment took place over a period of ten years from 2008 to 2018 and a total number of 2769 volunteers formerly exposed to asbestos were recruited in the MoMar study. All participants of the cohort gave written informed consent. The study was approved by the ethics committee of the Ruhr-University Bochum (reference number 3217-08).

Information about the participating individuals’ employment, former exposure to asbestos, imaging procedures, smoking status, diseases, and medication was recorded using questionnaires on every examination date.

The vast majority (99.3%) of the participants were men (N = 2749), whereas only 0.7% were women (N = 20). The median age of the participants was 73 years (with a range of 43–94 years). The basic characteristics of the MoMar cohort are depicted in Table 1.

Further information regarding the study population was published recently [18].

The recruitment started in December 2008. After four years of increasing numbers of newly recruited study subjects (for the first examination), the recruitment numbers leveled off in 2013, representing approximately 2000 participants (Figure 2). However, recruitment still continued over the whole study period. Similar growth curves could be observed for the following examinations, starting with an intentional spacing of one year between examinations. The last examinations and blood collections were completed in March 2018 (Figure 2).

In general, the participants visited the medical practices annually for examinations and blood collections with a median interval between examinations of 12.3 months (an interquartile range (IQR) of 11.7–13.3). However, the time interval between examinations was larger than two years for 213 (8%) individuals.

Based on the fact that mesothelioma is a rare disease, it was difficult to recruit a sufficient number of study subjects within the intended study time. This was in line with Patuleia et al., who stated that predicting a time frame for recruitment is challenging [19]. Particularly, for rare cancers, an expanded recruitment period might be necessary. Although the expected number of 2000 individuals was reached within four years, dropouts are an unavoidable component of long-term studies and the recruitment process had to be continued [19]. Thus, a higher number of initial participants had to be recruited to compensate for recruitment delays and possible dropouts during the study period. After doubling the original time frame to ten years, a total of 14,939 person-years were recorded, which was a 49% increase from the expected 10,000 person-years at the beginning of the study.

It is relevant to monitor the serial acquisition because participation is likely to be discontinued [19]. The majority of the study subjects were examined repeatedly. The required five annual examinations were conducted for 541 participants and another 998 participants were examined even more frequently, e.g., some participants were even examined ten times. In contrast, 324 participants were examined only once.

Regular follow-up surveys regarding vital status and target diseases were performed annually and a final follow-up survey was completed 17 months after the last blood collection, in order to obtain the data of all of the cancer cases that occurred up to one year after the last blood collection. As outlined in the recent publication on the MoMar cohort [18], there were 40 incident mesothelioma cases registered as of the reporting date in March 2019. As 2.27 mesothelioma cases per 100,000 individuals were expected in the general population, a risk of 17.6 (a 95% CI of 12.57–23.96) was assessed in the MoMar cohort [18]. Additionally, three prevalent mesothelioma cases were identified. Prevalent cases were defined as cases receiving the diagnosis before or within 100 days after the first examination [18].

Because only 1.4% of the participants developed mesothelioma, it is obvious that appropriate longitudinal studies should be very large as well as long lasting. This was also observed by Patuleia et al., who showed that even for breast cancer, which occurs relatively frequently, only 1–6% of women at high risk developed cancer [19].

In addition to the mesothelioma cases, 64 incident lung cancer cases were observed in the MoMar cohort. Based on expected 50.46 lung cancer cases per 100,000 individuals in the general population, a risk of 1.27 (a 95% CI of 0.99–1.62) was recorded [18]. This is similar to the odds ratio of 1.24 (a 95% CI of 1.18–1.31) assessed in a large pooled analysis of population-based case-control studies [20].

### 3.2. Sample Processing and Storage

The procedure of sample collection has to be designed to ensure the integrity of the analytes [9]. While serum needs to be additionally clotted at room temperature for a defined time (commonly > 30 min), the procedure to obtain plasma from blood can be initiated immediately. Thus, plasma was collected in this study and we chose the widely used EDTA as an anticoagulant. Centrifugation was performed at room temperature and 2000 xg, within 30 min after collection. To simplify the procedure, gel monovettes were used to enable the fast and easy separation of plasma and the cellular fraction by decanting. Whole blood, plasma, and the cellular fraction were immediately frozen at −20 °C in the participating study centers. The time points of the different processing steps were recorded using a sample run sheet. Additionally, any irregularities that occurred were noted. The collection and processing steps for blood samples are presented in Figure 3.

Blood samples have to be preprocessed immediately after collection, because the sample tubes contain billions of metabolically active cells and components of limited stability [15]. For this purpose, compliance with the SOPs is very important. Thus, the samples’ run sheets, including information about the time of blood collection, centrifugation to obtain the plasma, and time of freezing, were evaluated. Overall, more than 12,500 plasma samples were collected. The median time between the blood being drawn and centrifugation was 5 min (with an IQR of 2–10 min) and the median time between centrifugation and the freezing of the plasma samples was 23 min (with an IQR of 15–35 min). Thus, the majority of procedures were performed principally within the predefined time interval of 30 min. For comparison, in the majority of published clinical studies, plasma samples were separated within two hours [15]. However, it could not be ruled out that individual plasma samples were processed with delays. In such cases, delays were noted on the sample sheets for later consideration.

To assess the sample quality, the hemolysis grade was determined in 242 plasma samples randomly selected from different study centers by measuring the amount of free hemoglobin (Hb) in the plasma using a spectrophotometer [21,22]. Hemolysis is defined as free Hb concentrations > 0.3 ng/mL, reflecting the lysis of approximately 0.2% of all erythrocytes in the sample [16]. The vast majority (N = 240; 99.2%) of the analyzed samples showed Hb concentrations < 0.3 ng/mL. Increased Hb levels could only be observed in two samples (0.8%). Thus, the collected plasma samples were of sufficient quality for subsequent laboratory analyses. Nevertheless, regarding circulating miRNAs as potential biomarkers, it is necessary to analyze the dependence of every candidate miRNA from hemolysis prior to the assessment of biomarker performance [23].

The conducted quality controls indicate that the instructions for blood collection and the proceeding of samples were followed according to the predefined SOPs, resulting in feasible plasma samples for subsequent biomarker analyses. However, additional individual quality controls regarding specific biomarkers, e.g., the integrity of the RNA, might be meaningful.

Samples were frozen and intermediately stored at −20 °C at the collaborating study centers, because sufficient freezing conditions at lower temperatures are usually lacking at clinics and medical practices. Thus, the samples had to be transferred to storage conditions at lower temperatures as soon as they arrived at the central study center [9]. Accordingly, blood samples were regularly transferred from the participating study centers to the central study center. All samples were thawed once for subsequent aliquoting into 1 mL aliquots and frozen again at −80 °C for timely analyses and at < −150 °C for long-term storage. A single freeze/thaw cycle was allowed during the whole process because such a procedure is unavoidable in real-life settings. Ideally, the different aliquots of individual samples should be stored in different freezers or even sites in case of a failure of the freezers or facilities. In this case, only a single aliquot would be compromised instead of the complete sample set.

The intended workflow of the MoMar study was well executed, resulting in samples that appeared to be appropriate for biomarker analyses, as shown in recent studies analyzing proteins and miRNAs [24,25]. However, not every potential biomarker can be tested and some might be sensitive to preanalytical factors, like an extended time interval between blood collection and processing. These biomarkers would not be useful for routine clinical use, because in daily work, a delay in the processing of samples or other irregularities might be common. Appropriate biomarkers need to be robust to be implemented as a valuable tool in routine clinical use. Thus, every candidate biomarker must be analyzed regarding possible biological, preanalytical, and analytical factors. Analyses of preanalytical factors with an impact on the stability of calretinin and mesothelin, the two biomarkers validated in the MoMar cohort for the early detection of mesothelioma [25], were performed in the initial studies, revealing a fundamental robustness of both proteins regarding the ambient temperature and freeze/thaw cycles [26,27], making them appropriate for routine use. Calretinin and mesothelin were additionally analyzed in a study group of the general population without the target diseases, revealing renal dysfunction as an essential influencing factor for calretinin and mesothelin [28], confirming previous results for mesothelin [26,29]. Such information is of high importance for possible biomarker applications in the future, e.g., in screening programs, by taking kidney function and other factors into account.

The traceability of samples is an important aspect in biobanking [30,31], as well as in prospective, longitudinal studies with serial sample collection. The mislabeling of samples with the wrong participants’ information might affect the study’s integrity and lead to inaccurate conclusions [31]. Besides automatic and robotic procedures using barcodes to avoid transfer errors, genetic methods for sample authentication improve quality control [32]. A simple strategy to detecting sample identification errors is the analysis of human leukocyte antigen (HLA) alleles [31]. Thus, in the very rare case of a suspected sample being swapped for another within a study center, the central laboratory was able to perform an HLA analysis according to Rihs et al. [33] using the already collected samples from the corresponding patients as references to clearly identify the sample.

The long-term storage of samples and corresponding data in a biobank is an essential part of prospective, longitudinal studies. These studies are time- and cost-intensive and sample analyses are usually performed after the end of recruitment and the final follow-up. When new biomarker candidates are identified even years later, such analyses could be carried out. Appropriate samples and corresponding data deposited in an affiliated biobank can be selected and analyzed easily and timely without the need to recruit a new cohort, making the initial effort an investment in the future. However, ethics approval, comprehensive informed consent, and data protection rules must be considered at the beginning of the study.

### 3.3. Nested Case-Control Design

For the analyses of candidate biomarkers, a nested case-control study within the prospective design was performed, allowing for the retrospective evaluation of the biomarkers by comparing cases with matched controls from the same target population [8,34]. In a nested case-control study, cases of the target disease that occur within the defined cohort are identified, and for each case, a number of matched controls is selected from the individuals of the same cohort who did not develop the disease [35]. Such a nested case-control design has been used in other biomarker studies, for example, for the validation of candidate biomarkers to detect ovarian cancer in the Prostate, Lung, Colorectal, and Ovarian (PLCO) Cancer Screening Trial [34]. Based on the high number of recruited participants, it is impractical to assess every candidate biomarker in all available samples, because these analyses would be very time- and cost-intensive, and require many samples. Thus, it might be more meaningful to perform a nested case-control study, which, if appropriately matched, emulates the underlying cohort and enables unbiased risk estimates as in the full cohort [36].

For the analysis of biomarkers for early detection, the last collected blood sample prior to diagnosis was selected from each individual with a confirmed diagnosis of mesothelioma. There is evidence that only samples collected up to one year prior to diagnosis might show altered biomarker levels [34], whereas changes in samples collected years earlier might not be detectable.

In general, the selection of controls serving as an appropriate reference group is challenging when designing case-control studies. Ideally, controls should be taken from the same group from which the cases arise [37]. Thus, a major strength of nested case-control studies is that the controls are derived from the same cohort as the cases, resulting in an equal distribution of potential factors influencing the biomarkers [13]. Matching makes controls comparable to the available cases regarding subject- and sample-related factors. Typical matching criteria comprise age and gender as well as several other biological and preanalytical factors [8]. In the MoMar study, the controls were matched to cases using age, gender, and the time of blood collection (Figure 4).

At the conclusion of the follow-up, 43 mesothelioma (40 incident and 3 prevalent cases) cases were registered in the study that were matched to 172 controls, using a matching ratio of 1:4.

The planned time interval between examinations was one year, an interval that will also be applied in future screening programs. In reality, scheduling problems, delays caused by illness, or other unexpected events can lead to longer intervals. On the other hand, samples taken significantly more than one year or less than a few weeks before diagnosis are usually not very informative [38]. Therefore, only incident cases that had samples available in the time interval between 13 months and three weeks before diagnosis were included in the analyses. This resulted in 32 eligible cases for biomarker evaluation, with a median interval of 7 months (an IQR of 4–9 months) between blood collection and diagnosis of the disease. In the initial analysis, 26 cases were eligible regarding the chosen time interval [25]. For the update of the biomarker performance, all analyses were performed as described previously [25,39]. Detailed information is presented in the Appendix A.

The assessment of the performance of calretinin and mesothelin to detect mesothelioma prior to diagnosis resulted in a median calretinin level of 0.40 ng/mL (an IQR of 0.29–0.68 ng/mL) in prediagnostic mesothelioma cases and 0.18 ng/mL (an IQR of 0.12–0.29 ng/mL) in the controls, and a median mesothelin level of 1.43 nmol/L (an IQR of 0.97–2.66 nmol/L) in prediagnostic mesothelioma cases and 1.03 nmol/L (an IQR of 0.75–1.47 nmol/L) in the controls (Figure 5).

The concentrations of the biomarkers calretinin and mesothelin are presented in the Appendix A. Differences between prediagnostic mesothelioma cases and the controls were statistically significant for calretinin (*p* < 0.0001) and mesothelin (*p* = 0.0005).

The receiver operating characteristic (ROC) analysis resulted in an area under the curve (AUC) of 0.80 (95% CI 0.70–0.90) for calretinin, 0.69 (95% CI 0.59–0.80) for mesothelin, and 0.86 (range 0.86–0.68) for the combination of both biomarkers in the nested case-control study (Figure 6). For a sequential combination of the two markers, first calretinin was used for classification, and subsequently, mesothelin was examined only for calretinin-negative subjects.

Using the previous defined specificity of 98% [25], sensitivities of 34% for calretinin and 25% for mesothelin were revealed, whereas the combination of both biomarkers resulted in an increased sensitivity of 44%.

For a further improvement of the panel performance, additional candidate biomarkers were evaluated. As the inclusion of biomarkers of different molecular classes into the panel might be a reasonable approach, miRNAs were additionally validated. Based on the results in initial case-controls studies, three miRNAs, namely miR-103a-3p, miR-126, and miR-132-3p, were recently analyzed using a subset of a nested case-control study comprising 17 mesothelioma cases and 34 matched asbestos-exposed controls [24]. Unfortunately, all three analyzed miRNAs failed to detect mesothelioma in the prediagnostic samples taken 8.9 months (an IQR of 5.5–12.1 months) prior to the diagnosis of mesothelioma. Thus, the analyzed miRNAs are not appropriate complementary biomarkers for the early detection of mesothelioma [24].

## 4. Discussion

Prospective, longitudinal studies are large, long lasting, and cost-intensive. Thus, this study design contains a variety of challenges [40]. Nevertheless, for the assessment of the biomarkers’ performance for the early detection of rare diseases, this study design is essential.

In 2008, our prospective, longitudinal, multicenter MoMar study started in order to recruit participants formerly exposed to asbestos. The recruitment ended after about a decade and 2769 participants were enclosed in the cohort by 2018, representing 12,548 examinations in total. For the validation of candidate biomarkers, blood samples were collected from the participants annually, and for the assessment of the biomarkers’ performance, a nested case-control design was applied. The general advantages of a nested case-control design with archived samples of the cohort in a biobank are as follows: (i) cost reduction and effort minimization, because only parts of the complete sample collection and information have to be analyzed; (ii) reduced selection bias, because all cases and matched controls originate from the same cohort; and (iii) the possibility to analyze further biomarkers at a later date that were not considered at the beginning of the original study [41]. A disadvantage of a nested case-control study is its reduced power in comparison to that of the original complete cohort based on a smaller sample size [41]. But in general, the analyses in a nested case-control study within a prospective cohort should provide similar results as those in the complete cohort [13]. Using the nested case-control design, 26 mesothelioma cases and 136 matched controls were analyzed in the initial analysis in 2018 [25]. The biomarkers calretinin and mesothelin were successfully validated showing a sensitivity of 46% at a predefined specificity of 98% [25]. As the final follow-up was concluded in 2019, additional mesothelioma cases could be recorded, and finally, 43 mesothelioma cases were recorded in the MoMar study, of which 32 were eligible for analysis. The re-evaluation of the biomarkers’ performance confirmed the initial results with a sensitivity of 44% at a predefined specificity of 98%. Calretinin and mesothelin were the first biomarkers in combination to be validated for the early detection of mesothelioma using liquid biopsies. Previously, only mesothelin as a single biomarker had been evaluated in a prospective cohort [42]. In the future, these biomarkers have to be assessed regarding the benefits of early detection using biomarkers versus those of usual health surveillance. The ultimate goal would be proof of a reduction in mortality by the usage of biomarkers [8].

Initially, the MoMar study was intended to run for five years, but in the course of the study, the time frame had to be doubled. This led to a 50% increase of the initially estimated person-years and a sufficient number of incident mesothelioma cases. Overall, when performing a prospective, longitudinal study, one should be prepared for a prolonged runtime [19].

Continual follow-ups are an important part of prospective studies. However, it is essential to perform additional follow-ups after the end of the active field phase of the study, because the disclosure of information might be delayed. An additional follow-up might also identify new cases that have been missed before or might have just occurred later than one year after the last blood sample was taken. In regards to this, it might be interesting if some candidate biomarkers might be appropriate for the detection of the target disease even earlier than one year.

Using prediagnostic specimens of a cohort for the future discovery and validation of new biomarker requires correct sample handling, processing, and storage. Generally, in prospective, longitudinal long-term studies, the stability and integrity of samples until the analysis stage is a major problem [19]. In the MoMar study, all the plasma samples were immediately frozen after collection and the cold chain was maintained until the analysis. For the necessary aliquoting, a single freeze/thaw cycle was unavoidable because any additional processing is difficult to perform in routine clinical settings. A single freeze/thaw cycle is therefore common practice [9]. Thus, it should be taken into account that appropriate biomarkers for routine clinical use should be robust against unavoidable factors [8], particularly preanalytical factors like hemolysis, storage temperature, and freeze/thaw cycles. Appropriate SOPs have to be developed according to good laboratory practice in biomarker research, and it is necessary to follow these SOPs strictly.

It is not unusual that so-called interval cancers occur between regular examinations. For breast cancer, approximately 3–17% of the tumors are diagnosed between examinations [19]. In its final developmental stages, mesothelioma is a fast-growing cancer that might elude the screening intervals [43]. Thus, a suitable time window between examinations should be defined, which can be covered by appropriate biomarkers. Ideally, the length of the interval between examinations lies between the detectability and the incurability of the disease [8]. For calretinin and mesothelin, the appropriate interval to detect mesothelioma is about one year prior to the clinical diagnosis [25]. Analyzing several biomarkers for the early detection of ovarian cancer, Blyuss et al. assessed an improvement in detection up to 18 months prior to the clinical diagnosis [38]. Taking into account that the effort for participants of a surveillance program should be reasonable and keeping in mind that possible delays in scheduling might occur, it seemed justified to use a time interval between examinations of approximately one year. However, interval cancers can never be excluded.

As 64 incident lung cancer cases were recorded during the MoMar study, the cohort is also useful for the validation of biomarkers for the early detection of lung cancer which are urgently needed as well.

Besides the limitations presented recently [18], a basic limitation of this study is that only 20 (0.7%) of the 2769 participants were female and this number was too small to assess reliable information regarding women. Thus, the samples and data from female participants have to be excluded from the majority of analyses, although the annual number of deaths among women due to mesothelioma has increased [44]. In the future, it is necessary to establish a cohort of female participants formerly exposed to asbestos in order to validate candidate biomarkers in women. Another drawback of the study is that the localization and stage of the tumor remains unknown for the time point of the collection of each sample and could not be correlated to the corresponding biomarker concentrations, because repeated imaging of the asymptomatic participants was neither justifiable nor possible. Circulating biomarkers are not suitable to localize or stage tumors [45]. Thus, diagnostic procedures might consist of two parts: first, the measurement of circulating biomarkers to detect the cancer; and second, imaging techniques to localize and stage the tumor. The final diagnosis has to be confirmed by the cyto- or histopathological examination of materials obtained in subsequent invasive procedures.

## 5. Conclusions

The MoMar study represents a highly suitable prospective, longitudinal, multicenter study for the validation of biomarkers for the early detection of mesothelioma as well as lung cancer. The combination of calretinin and mesothelin was validated for the early detection of mesothelioma approximately one year prior to the common diagnosis based on clinical symptoms. An early detection of mesothelioma as well as other malignant diseases might be a promising approach to support options in therapy, because therapies at early stages might be more effective. As more than 12,500 plasma samples from 2769 individuals and corresponding data were stored within our affiliated biobank, it will be feasible to quickly validate other promising biomarkers in the future. In particular, in the context of lung cancer screening programs, the prediagnostic samples from 64 lung cancer cases might be of high interest for the validation of candidate markers for the early detection of lung cancer.

## Figures and Tables

**Figure 1 cancers-15-05896-f001:**
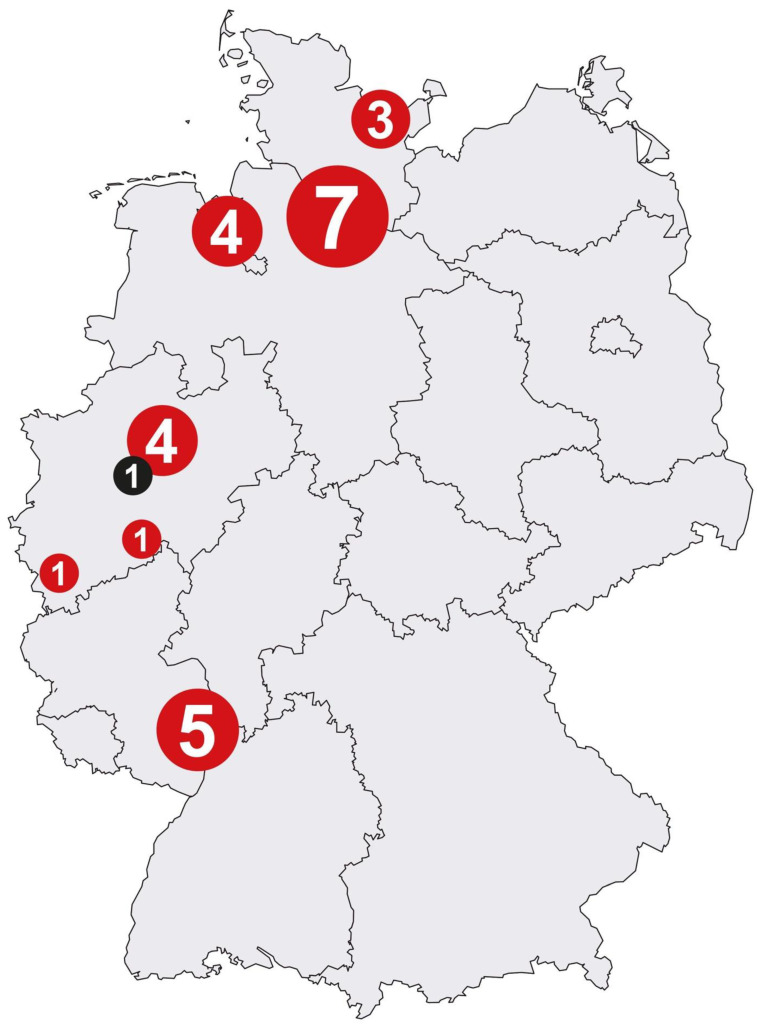
Localization of 26 collaborating study centers (red dots) and the central study center (black dot) in Germany. The number of participating study centers in the corresponding areas is indicated in the dots.

**Figure 2 cancers-15-05896-f002:**
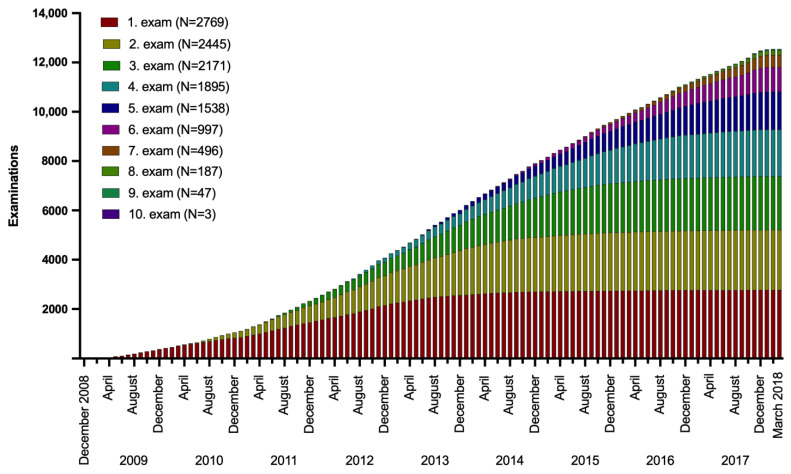
Courses of cumulative examinations during the study period of ten years between 2008 and 2018.

**Figure 3 cancers-15-05896-f003:**
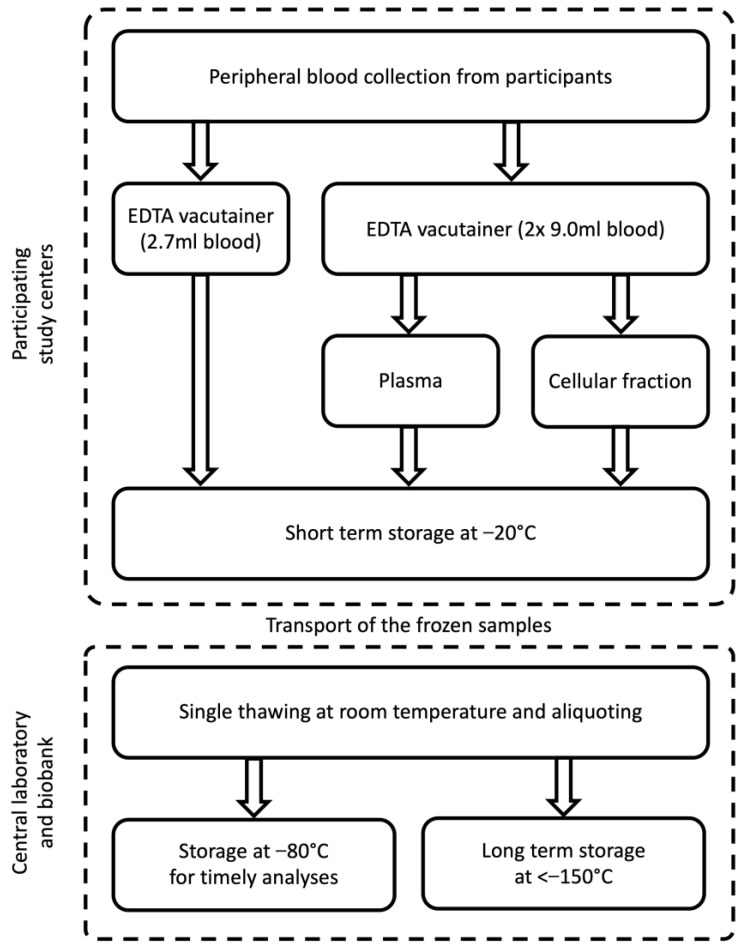
Collection and processing of blood samples in the MoMar study.

**Figure 4 cancers-15-05896-f004:**
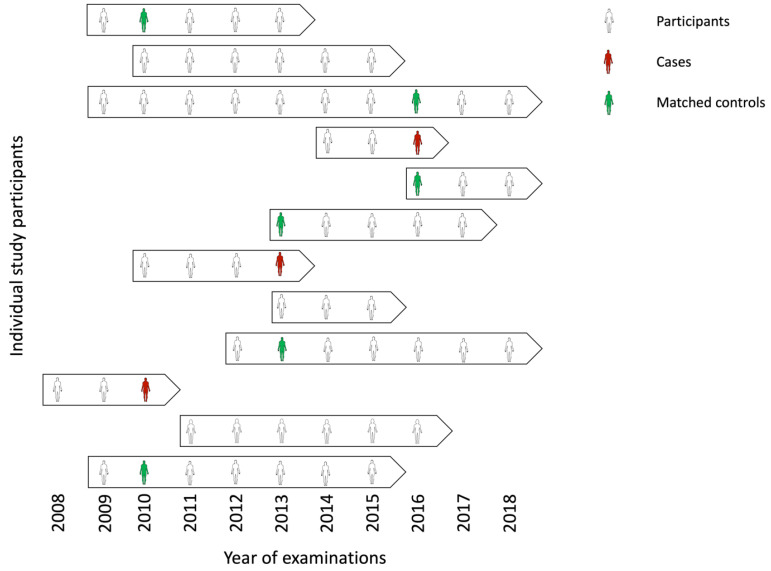
Annually examinations of the participants in the prospective MoMar study. Cases and matched controls for the nested case-control study originate from the pool of the participants. Matching was performed based on age, gender, and date of blood collection. Each arrow represents an example of repeated examinations of a single individual over the indicated years. The starting point of the arrows indicates the year of enrollment in the study and the end point indicates the exit of the subjects from the study.

**Figure 5 cancers-15-05896-f005:**
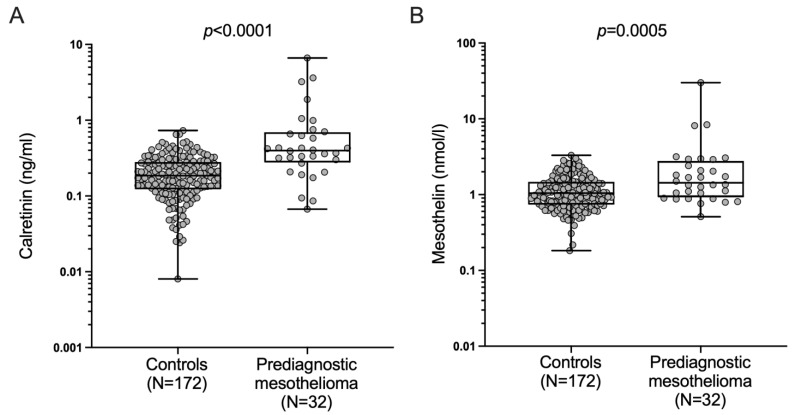
Distribution of (**A**) calretinin and (**B**) mesothelin concentrations in 32 prediagnostic mesothelioma cases and 172 asbestos-exposed controls. Circles represent single biomarker values. Boxes represent 25th percentile, median, and 75th percentile and whiskers represent the minimum and maximum.

**Figure 6 cancers-15-05896-f006:**
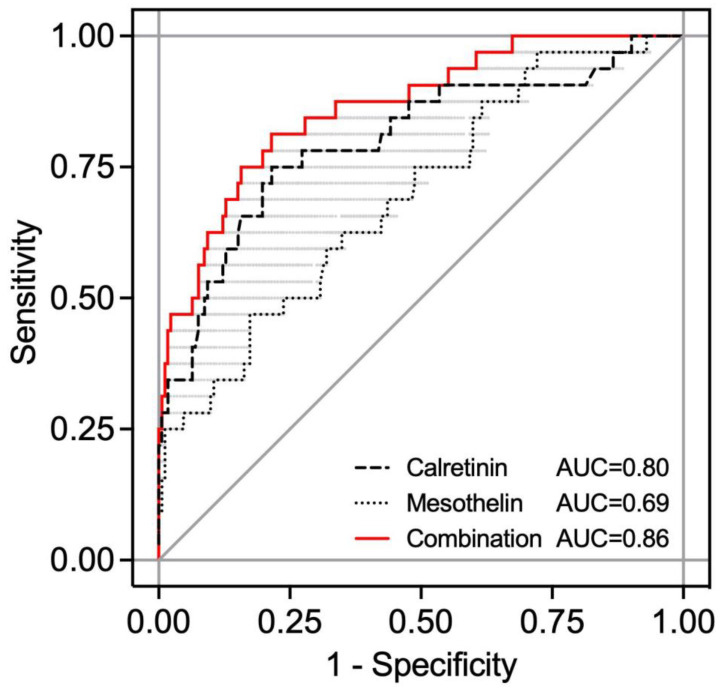
Receiver operating characteristic (ROC) analysis of calretinin, mesothelin, and the combination of both biomarkers. ROC curves and area under curve (AUC) values of calretinin, mesothelin, and the combination of both biomarkers based on 32 prediagnostic mesothelioma cases and 172 controls. The combination is based on a “best case” sequential combination.

**Table 1 cancers-15-05896-t001:** Basic characteristics of the study population.

Characteristics	
Total [n]	2769
Gender [n (%)]	
Male	2749 (99.3)
Female	20 (0.7)
Age in years at study entry [median (range)]	73 (43–94)
Smoking status [n (%)]	
Never smokers	763 (27.6)
Former smokers	1696 (61.2)
Current smokers	304 (11.0)
Unknown	6 (0.2)
Duration of asbestos exposure in years [median (range)]	27 (1–60) *
Time in years between last exposure and study entry [median (range)]	23 (1–68) *

* Based on 2652 participants, because exposure information was not available for 117 participants.

## Data Availability

The Appendix A contains the subjects’ characteristics and the concentrations of the biomarkers calretinin (ng/mL) and mesothelin (nM).

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
