# Peer review of "Tasks and Experiences of the Prospective, Longitudinal, Multicenter MoMar (Molecular Markers) Study for the Early Detection of Mesothelioma in Individuals Formerly Exposed to Asbestos Using Liquid Biopsies"

_cancers, 2023, doi:10.3390/cancers15245896_

Round 1

Reviewer 1 Report

Comments and Suggestions for Authors

The article of Weber and colleagues is an exhaustive and well written overview of the author experience in establishing a blood biobank for cancer monitoring, in their case pleural mesothelioma. This will be informative and of help for any researchers who intend to work in a similar direction or making use of biobank specimens. As minor points, I ‘d suggest some points that may profit of additional clarifications and contextualization:

-        What kind of data/information are collected in the yearly visits (e.g. also radiology?) and how is this reflected in figure 2

-        Not sure how to interpret figure 4: what would be reported on the y-axis, are these the different blocks? And what are the blocks (“arrow”)?

-        It would be helpful to very briefly explain what nested case-control study is before its justification and to discuss what other method could have been used in alternative

-        Figure 5: please explain what the boxed and lines are (whiskers are explained)

-        Please explain how were calretinin and mesothelin combined (best case sequential combinations)

-        Within the cohort there were 40 mesothelioma cases and 61 lung cancers. Would have not been natural to expect a higher number of lung cancer cases compared to mesothelioma considering that lung cancer is more frequent and that asbestos is a risk factor for lung cancer as well and many of the individuals of the cohort were smokers? At what timepoint of the screening were the lung cancer cases detected (after how man years from enrolment)? That would be an interesting information in the context of growing lung cancer screening programs.

Author Response

The article of Weber and colleagues is an exhaustive and well written overview of the author experience in establishing a blood biobank for cancer monitoring, in their case pleural mesothelioma. This will be informative and of help for any researchers who intend to work in a similar direction or making use of biobank specimens. As minor points, I ‘d suggest some points that may profit of additional clarifications and contextualization:

We thank the reviewer for the valuable comments. We revised our manuscript and briefly address each point.

-What kind of data/information are collected in the yearly visits (e.g. also radiology?) and how is this reflected in figure 2

Information of the participating individuals regarding job, former asbestos exposure, imaging procedures, smoking status, diseases, and medication were recorded at every examination date. This is now stated in lines 211 - 213. However, this information was not reflected in Figure 2, the figure represents the cumulative number of examinations in each year.

-Not sure how to interpret figure 4: what would be reported on the y-axis, are these the different blocks? And what are the blocks (“arrow”)?

You are correct, blocks represent the individual duration of the subjects in the study as examples. Each arrow represents the repeated examinations of a single individual over the indicated years. The starting point of the arrows indicates the year of enrollment in the study and the end point indicates the exit of the subjects from the study. This is now clarified in the corresponding figure legend (lines 412 - 414).

-It would be helpful to very briefly explain what nested case-control study is before its justification and to discuss what other method could have been used in alternative

Thanks for your suggestion. In a nested case-control study, cases of the target disease that occur within the defined cohort are identified and for each case a number of matched controls is selected from the individuals of the cohort who did not develop the disease. This definition is now stated in lines 381 - 384 and the reference (Ernster, 1994, Nested case-control studies) is added. An alternative method would have been a case-cohort design, where controls are selected as a subcohort at the beginning of the follow-up period and can be used as common control group for different outcomes. However, our cohort is not suited to analyze further diseases and a case-cohort design was not beneficial to us.

-Figure 5: please explain what the boxed and lines are (whiskers are explained)

Boxes represent median, 25th, and 75th percentile. This is now added in the corresponding figure legend (lines 443 - 444).

-Please explain how were calretinin and mesothelin combined (best case sequential combinations)

For the sequential combination of the two markers, first calretinin was used for classification. And subsequently, mesothelin was examined only for calretinin-negative subjects. This is now described in lines 452 - 454.

-Within the cohort there were 40 mesothelioma cases and 61 lung cancers. Would have not been natural to expect a higher number of lung cancer cases compared to mesothelioma considering that lung cancer is more frequent and that asbestos is a risk factor for lung cancer as well and many of the individuals of the cohort were smokers? At what timepoint of the screening were the lung cancer cases detected (after how man years from enrolment)? That would be an interesting information in the context of growing lung cancer screening programs.

In the MoMar cohort 50.46 lung cancer cases were expected corresponding to a standard incidence ratio (SIR) of 1.27 (95% CI: 0.99-1.62) as recently described (Taeger et al., 2022. Lung cancer and mesothelioma risks in a prospective cohort of workers with asbestos‐related lung or pleural diseases). Additionally, 90% of the lung cancer cases had ever smoked compared to 70% of controls and mesothelioma cases. A stratified analysis by smoking status showed that the never smokers had no excess lung cancer risk (SIR=0.46, 95% CI: 0.17-1.01), whereas former and current smokers showed elevated lung cancer risks (SIR=1.40, 95%CI: 1.03-1.87 for former smokers; SIR=2.70, 95% CI: 1.35-4.83 for current smokers). Therefore, in our study we attributed lung cancer risk to the increased smoking prevalence. It may also mask the asbestos-related risk in our cohort. Lung cancer was not detected using biomarkers, because no reliable biomarkers for lung cancer are validated yet. Instead, the lung cancer cases were diagnosed during clinical routine. Thank you for your note, that in the context of lung cancer screening programs the prediagnostic samples might be of high interest for validation of candidate markers for the early detection of lung cancer. This is now added in the Conclusions section (lines 580 - 582).

Reviewer 2 Report

Comments and Suggestions for Authors

The article has a highly detailed procedure of the trial as if it were an SOP, but no original results are shown that could support the discussion and conclusion.

The objective of the paper is not appropriate for an original article.

There are already documents from several societies that determine the minimum procedures for using liquid biopsy, and it would be optional to have so much text describing these variables.

The methodology described in the literature in citation number 25 (Johnen et al., 2018) must be more original to support this publication. It measures the protein markers calretinin and mesothelin, which were determined by enzyme-linked immunosorbent assays in prediagnostic plasma. The supplementary doc (S1) only describes the use of commercial kits already validated for these markers.

These markers have been highly studied and described in the literature. The results do not differentiate the values according to histological subtypes of mesothelioma defined in the work, associating them with differential survival.

Author Response

The article has a highly detailed procedure of the trial as if it were an SOP, but no original results are shown that could support the discussion and conclusion.

The objective of the paper is not appropriate for an original article.

There are already documents from several societies that determine the minimum procedures for using liquid biopsy, and it would be optional to have so much text describing these variables.

The methodology described in the literature in citation number 25 (Johnen et al., 2018) must be more original to support this publication. It measures the protein markers calretinin and mesothelin, which were determined by enzyme-linked immunosorbent assays in prediagnostic plasma. The supplementary doc (S1) only describes the use of commercial kits already validated for these markers.

These markers have been highly studied and described in the literature. The results do not differentiate the values according to histological subtypes of mesothelioma defined in the work, associating them with differential survival.

We thank the reviewer and take note of the points raised. However, the aim of the manuscript is to provide a detailed overview of lessons learned and experiences made conducting a prospective, longitudinal, multicenter study which is elementary to validate candidate biomarkers for the early detection of cancer. This is already stated in the abstract (lines 32 - 33) and introduction (lines 107 - 110). So far, such longitudinal studies for mesothelioma are extremely rare. The reviewer is right in pointing out the problem of different mesothelioma subtypes and corresponding overall survical rates. However, the detection of non-epithelioid mesothelioma remains more difficult than for epithelioid mesothelioma. This might be due to variations in the release of biomarkers from different histological subtypes into the bloodstream. However, new approaches in therapy using the immune checkpoint inhibitors nivolumab and ipilimumab showed an improved benefit especially for non-epithelioid mesothelioma. The corresponding information is already stated in the introduction (lines 60 - 65).

Reviewer 3 Report

Comments and Suggestions for Authors

The current manuscript reports a prospective, longitudinal, and multi-center study for the early detection of malignant mesothelioma in individuals formerly exposed to asbestos using liquid biopsies. The study recruited participants at risk for over 10 years and allowed for 17 months of follow up after last blood draw. Authors share their experiences and lessons learned from the study. The study validated the combination of calretinin and mesothelin for the early detection of mesothelioma by approximately one year before the onset of the disease. They also tested three miRNAs and showed that none of them have ability to detect mesothelioma before clinical diagnosis.

Early detection of cancer is fundamental for a successful treatment. Using liquid biopsy for the early detection of cancer is a hot topic as it is a minimally invasive method and allows for longitudinal monitoring of cancer. The current study is well designed and executed and I thank authors for sharing their experience with the scientific community. I have no major concerns, apart from few typos. One important typo is the number format of the incidence rates of mesothelioma in Germany. In the introduction section authors stated “ In Germany, on average 1.279 men and 337 women were diagnosed with mesothelioma annually between 2010 and 2019 (1).” I believe the 1.279 should be written “1,279”.

Comments on the Quality of English Language

English is good but there few typos which I think will be spotted in the proofreading

Author Response

The current manuscript reports a prospective, longitudinal, and multi-center study for the early detection of malignant mesothelioma in individuals formerly exposed to asbestos using liquid biopsies. The study recruited participants at risk for over 10 years and allowed for 17 months of follow up after last blood draw. Authors share their experiences and lessons learned from the study. The study validated the combination of calretinin and mesothelin for the early detection of mesothelioma by approximately one year before the onset of the disease. They also tested three miRNAs and showed that none of them have ability to detect mesothelioma before clinical diagnosis.

Early detection of cancer is fundamental for a successful treatment. Using liquid biopsy for the early detection of cancer is a hot topic as it is a minimally invasive method and allows for longitudinal monitoring of cancer. The current study is well designed and executed and I thank authors for sharing their experience with the scientific community. I have no major concerns, apart from few typos. One important typo is the number format of the incidence rates of mesothelioma in Germany. In the introduction section authors stated “ In Germany, on average 1.279 men and 337 women were diagnosed with mesothelioma annually between 2010 and 2019 (1).” I believe the 1.279 should be written “1,279”.

We thank the reviewer for this positive assessment. The manuscript was proofread and typos in the text and figures were corrected. Additionally, the prefix “malignant” has been omitted from mesothelioma, because all mesothelioma cases are regarded as malignant, according to Sauter et al., 2021, The 2021 WHO classification of Tumors of the Pleura: Advances since the 2015 Classification.

Reviewer 4 Report

Comments and Suggestions for Authors

I have no comments or suggestions: the MoMar study has been described in detail, as well as its main findings 

Author Response

I have no comments or suggestions: the MoMar study has been described in detail, as well as its main findings.

We thank the reviewer for the positive evaluation of our manuscript.